# Thrips as the Transmission Bottleneck for Mixed Infection of Two *Orthotospoviruses*

**DOI:** 10.3390/plants9040509

**Published:** 2020-04-15

**Authors:** Kaixi Zhao, Cristina Rosa

**Affiliations:** Department of Plant Pathology and Environmental Microbiology, The Pennsylvania State University, University Park, PA 16802, USA; kxz145@psu.edu

**Keywords:** orthotospovirus, mixed infection, TSWV, INSV, thrips, vector preference, bottleneck

## Abstract

Mixed infections provide opportunities for viruses to increase genetic diversity by facilitating genomic reassortment or recombination, and they may lead to the emergence of new virus species. Mixed infections of two economically important orthotospoviruses, *Tomato spotted wilt*
*orthotospovirus* (TSWV) and *Impatiens necrotic spot orthotospovirus* (INSV), were found in recent years, but no natural reassortants between INSV and TSWV were ever reported. The goal of this study was to establish how vector preferences and the ability to transmit INSV and TSWV influence transmission and establishment of mixed infections. Our results demonstrate that thrips prefer to oviposit on TSWV and INSV mixed-infected plants over singly infected or healthy plants, providing young nymphs with the opportunity to acquire both viruses. Conversely, we observed that thrips served as a bottleneck during transmission and favored transmission of one of the two viruses over the second one, or over transmission of both viruses simultaneously. This constraint was relaxed in plants, when transmission of TSWV and INSV occurred sequentially, demonstrating that plants serve as orthotospovirus permissive hosts, while thrips serve as a bottleneck. Viral fitness, as measured by virus replication, transmission, and competition with other viral strains, is not well studied in mixed infection. Our study looks at the success of transmission during mixed infection of orthotopoviruses, enhancing the understanding of orthotospovirus epidemiology and evolution.

## 1. Introduction

Orthotospoviruses are thrip-transmitted viruses that cause important economic losses to agriculture worldwide. *Tomato spotted wilt orthotospovirus* (TSWV), the type species of the *Orthotospovirus* genus, can infect more than 900 species of plants belonging to more than 90 families [1]. *Impatiens necrotic spot orthotospovirus* (INSV), initially designated as the TSWV-I [2] strain, was later upgraded to an independent species [3]. Compared with TSWV, INSV has a relatively narrow reported host range that includes about 300 species in 85 plant families [1].

Most plant viruses are transmitted by insects, including orthotospoviruses. While TSWV and INSV are generalists with regard to their plant hosts, both viruses and all other virus species in the genus have a narrow invertebrate vector range. In fact, until 2002, only five of the 160 described species of *Frankliniella* were known to be vectors and hosts of orthotospoviruses, only three of the 280 species from the genus *Thrips*, and one out of 90 species of *Scirtothrips* [4]. Up to 2015, 15 insect species in total were reported to serve as vectors and hosts of orthotospoviruses [5], nine thrips species as a vector of TSWV, and four of them as a vector of INSV [6,7,8,9].

Orthotospoviruses are transmitted in a persistent, propagative manner by their vectors, making them one of the few examples of viruses that can infect hosts from different kingdoms (Animalia and Plantae). The relationship between thrips and TSWV is unique, due to the fact that only the first-and second-stage larvae can acquire the virus. The site of egg hatching is particularly important. Larvae are wingless and preferentially feed on the plant on which they hatched. After being ingested by thrips, TSWV encounters multiple barriers in different tissues that serve as a bottleneck for virus transmission. Adult thrips cannot acquire TSWV even by feeding on infected plants for an extended period of time. Thrips lose the ability to acquire TSWV along with age; the first instar larvae have higher acquisition efficiency than the second instar larvae [10]. Our knowledge of the transmission of most orthotopsoviruses is based on what we know about TSWV transmission and is considered not to differ substantially from that mentioned above.

Two viruses can be inoculated into the same plant simultaneously or sequentially, resulting in mixed infections, a phenomenon that occurs often in nature. Viral synergy was first described in the 1950s with mixed infection of the potexvirus *Potato virus X* (PVX) and the potyvirus *Potato virus Y* (PVY). Mixed infection of those viruses resulted in a markedly more severe disease than infections caused by either virus alone. The beneficiary virus, which in this case was PVX, could reach a higher titer in mixed compared to single infection, whereas PVY accumulation remained unaltered [11]. While unrelated viruses generally interact in a synergistic way, the interactions between related viruses are mostly antagonistic. In the past several decades, antagonistic relationships, also known as super-infection exclusion (SIE), were used as a form of cross-protection [12]. In cross-protection, growers inoculate a mild isolate of a virus to protect plants against more severe strains of the same virus. SIE, the natural occurrence of cross-protection, is defined as the phenomenon in which infection by a virus excludes a second virus from entering already infected tissues.

Mixed infections of TSWV and INSV were first reported in tomato plants in Italy in 2000, and many new cases were reported in recent years [13,14]. Orthotospoviruses can also infect weeds and other non-crop plants where thrips can overwinter and reproduce. A survey conducted throughout Georgia in 2003 determined that more than 15 species of weed were positive for INSV, TSWV, or INSV and TSWV [14]. That survey was prompted by the occurrence of outbreaks of TWSV and INSV in tobacco fields adjacent to the infected weeds and finding that the infected tobacco was either infected by TSWV or by both viruses, but not by INSV alone. This study suggested that those outbreaks could have originated from migration of infected thrips from the surrounding weeds and that, aside from in planta interactions between the two viruses and between each virus and its host plant, vector transmission is involved in TSWV and INSV disease prevalence in the agricultural and natural landscapes.

In order to be efficiently transmitted from one plant to another, plant viruses induce chemical and physical changes in their plant hosts, aimed at attracting or repelling vectors [15,16,17]. In the case of thrips and TSWV, thrips feeding alone induces jasmonate (JA)-regulated defenses in plants [18,19], which negatively affects the performance and preference of thrips. However, TSWV infection increases infected plant salicylic acid (SA) content and induces SA-regulated gene expression [20]. By upregulating the SA pathway, TSWV infection de facto favors thrips feeding and fitness in *Arabidopsis thaliana* by decreasing the level of JA-regulated gene expression induced by thrips feeding [20]. Moreover, the non-structural proteins (NSs) encoded by TSWV can suppress plant volatile monoterpene biosynthesis via direct interaction with the JA pathway master regulator MYC2 and its two homologs MYC3 and MYC4 [21]. MYCs mediate the activation of terpene synthase genes [22,23], and monoterpenes were found to be repellent for western flower thrips [21]. Therefore, plants infected with TSWV are more attractive for *F. occidentalis* and uninfected females prefer infected plants for feeding and oviposition [24,25].

Virus infection can also alter vector feeding behavior to enhance transmission [26,27]. TSWV modifies the vector *F. occidentalis*’ behavior to benefit virus transmission. TSWV-infected male thrips probe three times more than uninfected males [26]. Virus transmission is recognized as an important component that, together with replication and epidemiological fitness, contributes to the total fitness of viruses [28]. Based on these studies and on the concept of transmission as a measure of fitness, it is logical to speculate that thrips’ preferences and transmission efficiency could play a role in maintaining or preventing mixed infections and change overall viral fitness.

Genetic diversity is thought to be important in allowing viruses to effectively survive within the host, and in facilitating adaptation to new hosts and changing environments [29]. Mixed infections provide viruses with a shared space where they can increase genetic diversity through recombination or reassortment [30,31], resulting in the emergence of new virus species.

Reassortment of RNA segments (genetic shift) with accumulation of point mutations is a powerful mechanism for bunyavirus evolution [30]. Many recognized viruses in the *Bunyavirales* were shown to be reassortants. If two related bunyaviruses share an ecologic niche, then the reassortment is more likely to happen. Rodriguez and McElroy’s studies suggest that orthohantaviruses can reassort when they are closely related and are able to co-exist in susceptible host cells [32,33].

Reassortment between two related viruses is well documented in other families in the order *Bunyavirales*; however, in the family *Tospoviridae*, it was first reported in 2011 [34]. *Groundnut ringspot orthotospovirus* (GRSV) and *Tomato chlorotic spot orthotospovirus* (TCSV) were first reported in the 1990s and are two emerging orthotospoviruses frequently detected in Solanaceae in Florida and the southeastern United States [35]. Since TSWV is already established in those states, the authors suggested that those states might become a location where new reassortments between TCSV, GRSV, and TSWV may occur. Reassortment between different TSWV strains facilitates rapid adaptation to new host genotypes [31], and sequence analysis of INSV isolates suggested that reassortment is the major driving force for INSV evolution [36]; thus, it is logical to hypothesize that co-infection of INSV and TSWV could lead to reassortment between the two viruses and to an increase in their host range.

Virus diversity within mixed-infected plants may reach a very high level and provide a chance for reassortment, but viruses are frequently subjected to bottleneck when they spread between hosts [37]. This could potentially eliminate mixed infection in the new plant host and decrease probability of reassortment. Many studies suggest that bottlenecks are severe during transmission, limiting the number of transmitted virions in a stochastic model, where the vectors can transmit a limited number of virus particles [38]. However, there are some exceptions; bottlenecks could be relaxed for viruses sharing similar transmission mode, when, for instance, two virus populations can complement each other, or when the two populations have a high recombination rate [38,39,40]. When the transmission bottleneck is relaxed, transmission does not significantly impact genetic diversity in the new host, and the new virus quasispecies will be more similar to the one in the original source [40].

Since nothing was known about how mixed infections of TSWV and INSV are established and maintained in agriculture, the goal of this study was to establish if TSWV and INSV could start successful mixed infections in their plant hosts and to see (1) if vector preference could be responsible for disease prevalence in crops, and (2) if thrips could be the transmission bottleneck for mixed infection of orthotospoviruses. Our results demonstrate that INSV and TSWV can co-infect plants, and that thrips prefer to lay eggs in mixed-infected leaves over singly infected or healthy leaves. While we could not pinpoint the source of attraction of thrips to mixed-infected plants, our data suggest that attraction to mixed-infected plants over single virus-infected plants could be due to a combination of plant volatiles with plant defense compounds induced by virus infection and thrips feeding. Furthermore, our results provide evidence that, while thrips represent a bottleneck for co-transmission of TSWV and INSV, mixed infections of these two viruses in plants can easily occur through sequential virus transmission. The permissive nature to mixed infection of plants vs. thrips suggests that plants are the ideal host for the generation of reassortants in orthotospoviruses.

## 2. Results

### 2.1. Frankliniella occidentalis Oviposits Preferentially on INSV and TSWV Mixed-Infected Leaves

In all choice tests (Figure 1), the number of eggs deposited was significantly higher in the leaf disc mixed-infected with TSWV and INSV when female thrips were allowed to choose between mixed-infected leaf discs and singly infected (INSV or TSWV) or uninfected leaf discs of *Emilia sonchifolia*. *E. sonchifolia* is the host commonly used to maintain TSWV and is generally liked by *F. occidentalis*. Our result demonstrates that female thrips prefer mixed-infected leaves over singly infected or healthy leaves for their offspring. Results from the preference test suggest that vector thrips’ preference for mixed-infected plants may lead to an increase in transmission rate of TWSV and INSV mixed infection.

Since symptoms of TSWV and INSV in *E. sonchifolia* mixed-infected plants were indistinguishable from the ones seen in single virus infection (Figure 2), we tested if other cues rather than visual were involved in thrips behavior and compared volatiles from mixed-infected *E. sonchifolia* with those from singly infected or healthy plants using gas chromatography–mass spectrometry (GC–MS). In our results, we found a total of 42 volatiles identifiable with confidence after removal of common GC–MS contaminants (Appendix A).

Although we were unable to detect specific volatile compounds that only associated with mixed-infected plants, and although the profile of all plant volatiles collected from different treatments was similar based on Principal Component Analysis (PCA) (Appendix A), few volatiles already seen to be important in virus–plant–vector interactions appeared in our analyses (Appendix A). Among them, (*Z*)-3-hexen-1-ol acetate, already shown to attract the parasitic wasp *Cotesia glomerata* [41], was seen only in mixed-infected plants (two out of four mixed-infected plants), while undecanal (aldehydes) and d-limonene (monoterpenes) were not present in TSWV-infected plants but were present in the other treatments (in three healthy, two mixed-infected, and one INSV-infected plants out of four plants for each treatment; and in three healthy, one mixed-infected, and one INSV-infected plants out of four plants for each treatment). d-Limonene is a known attractant of some insects such as *Diaphorina citri* (also known as Asian citrus psyllid, the vector of citrus greening) and of their predators [42,43]. *Propylaea japonica* is more attracted to healthy sour orange (*Citrus aurantium*) because of the high level of the combination of d-limonene and beta-ocimene, and lack of methyl salicylate emitted by those plants [43]. Finally, citrus varieties that are tolerant to *Candidatus* Liberibacter asiaticus (the causal agent of citrus greening) emitted more aldehydes (undecanal, geranial, citronella, and neral) and monoterpenes including d-limonene [44].

On the contrary, methyl salicylate, a well-known SA-derived volatile that attracts the aphid predator *Coccinella septempunctata* [45] and the beneficial insect *Chrysopa nigricornis* [46], but is repellant to predators of *Diaphorina citri* [43], was only present in TSWV-infected plants (two out of four TSWV-infected plants).

### 2.2. INSV Is Preferentially Acquired and Transmitted from Singly and Mixed-Infected Plants

Results from the egg oviposition experiment suggested that the vector *F. occidentalis* prefers mixed-infected plants and this may lead to an increase in transmission rate of mixed infections. Thus, the ability of *F. occidentalis* to acquire and transmit TSWV, INSV, and both viruses was tested. The status of the infected plants at the time of acquisition was tested by ELISA, and plants with similar level of each virus (Optical Density O.D. = 1.7) were used for virus acquisition and transmission tests.

Virus retention by 21 groups of five thrips each from TSWV-, INSV-, or mixed-infected *E. sonchifolia* was evaluated by testing the presence of TSWV and INSV by RT-PCR in the insect bodies two days after they reached adulthood. Seventeen out of 21 thrips groups were able to retain only INSV from mixed-infected plants, one group retained both TSWV and INSV (total of 18 groups for INSV), and one group retained only TSWV from mixed-infected plants (for a total of two groups able to acquire TSWV). Two groups of thrips were unable to retain either virus. The virus retention rate for single thrips, calculated using binGroup [47], was as follows: 0.3078 (30.78 individuals in 100) for INSV, 0.0194 for TSWV, and 0.0095 for both viruses.

Experiment results of single thrips transmission using *E*. *sonchifolia* leaf discs showed that thrips tended to transmit TSWV (average 16.65%, *n* = 64) less efficiently compared with INSV (average 30.11%, *n* = 56), when insects were left feeding on either TSWV or INSV singly infected leaf discs (Figure 3). Both viruses could be transmitted from mixed-infected plants (average 3.37% for TSWV and 27.97% for INSV, *n* = 89). There was no significant difference in transmission efficiency for INSV between singly (average 30.11%) and mixed-infected (average 27.97%) plants. However, transmission efficiency for TSWV from mixed-infected plants was much lower (3.37%, *n* = 89) than from TSWV singly infected plants (16.65%, *n* = 64) (*p* < 0.05). No simultaneous transmission of TSWV and INSV was ever detected in this experiment.

Our results suggest that, even if TSWV retention and transmission efficiency were decreased by the presence of INSV, TSWV could still be transmitted to a new plant host, either by itself or, at a very low rate, together with INSV.

## 3. Discussion

As plants are not mobile, plant viruses rely on other organisms to be transmitted to the new hosts. Most plant viruses use insects as vectors [10], and they can often modify insects’ behavior to enhance their transmission rate [15,16,17,26,27]. Previous studies showed that TSWV-infected plants are more attractive to *F. occidentalis*, and uninfected females prefer infected plants for feeding and oviposition [25]. *F. occidentalis* can use both visual and olfactory cues to select plants [48]. In our experiments, mixed-infected *Emilia sonchifolia* plants displayed similar symptoms compared with single virus-infected plants (Figure 2), but uninfected *F. occidentalis* adult females preferred to oviposit on mixed-infected leaf discs over singly infected leaf discs (Figure 1). A note of caution for the interpretation of these results is that we did not access the success of hatching or the survival in the progeny. Other studies [25,26] showed that thrips thrive on TSWV-infected plants, but we are not aware of studies where the authors looked at thrips progeny success in mixed-infected plants.

Our study suggested that adult female thrips could use olfactory cues to select plants for their offspring. Solid-phase microextraction (SPME) is considered to be a useful tool to get a realistic picture of the volatiles emitted from fresh tomato [49], and it was used to study volatiles from TSWV-infected plants [49,50,51]. We used SPME fibers to collect volatile compounds from mixed-infected plants and compared the collected compounds with those emitted by healthy or single virus-infected plants. The differential presence of key volatiles recognized as repellant or attractant by thrips and other insects, such as d-limonene and methyl salicylate, supports the hypothesis that thrips attraction could be due to a combination of plant volatile compounds and plant defense compounds that were induced by thrips feeding and different virus infections. In fact, the horizontal (PC1) and vertical axis (PC2) of our PCA analysis showed that the combination of volatiles detected could explain less than 50% of variation. There are no data on *E. sonchifolia* plant volatiles or chemical defenses, complicating the analyses of our data and further experiments.

Because of its simplicity, the use of leaf discs instead of whole plants in TSWV transmission tests was extensively implemented by Wijkamp et al. starting in 1993 [52], and a comparison between the two systems was performed and discussed by the same author in 1996 [53]. Since then, leaf disc assays have been extensively used in multiple studies on thrips and orthotospovirus interactions [54,55,56,57], and they have been successfully applied to choice tests. A drawback of using leaf discs is the elicitation of plant responses linked to a wounding signal, probably a similar but much stronger signal than that elicited during thrips feeding [18,19,20]. In Reference [20], the authors showed that the play between hormonal pathways in plants is responsible for plant, vector, and orthotospovirus interactions. Based on these considerations, we cannot rule out that the choice made by the thrips used in this study was also influenced by wounding, in addition to virus infection.

Even if thrips preference can contribute to transmission, during transmission, not all viruses will be passed into the new host for replication, and the mode of transmission could influence the population that will be transmitted into the next round. Orthotospoviruses are transmitted by thrips in a persistent, propagative manner and can replicate in both plant hosts and thrips vectors. Bunyaviruses, as well as reoviruses and rhabdoviruses that are transmitted in a propagative manner, can replicate in organisms belonging to two kingdoms [58], and they are hypothesized to have originated from animal-infecting viruses that acquired the ability to infect plants via the acquisition of “movement proteins”, the key proteins necessary for virus cell-to-cell movement in plants [59]. This evolutionary history and the longer association of thrips and orthotospoviruses [10] could explain why a narrower bottleneck is constituted by thrips transmission, as well as the fact that viruses need to pass more barriers in insects to arrive to the salivary glands than in plants, where each cell is equally available for acquisition and transmission. Interestingly, TSWV enters through the thrips epithelial cells of the midgut anterior region [60], but how virions reach the salivary glands remains unknown [58].

Viruses that have a similar transmission mode could encounter a less strict bottleneck during transmission [40]. However, when mixed infections happen between closely related viruses, the transmission bottleneck may not be loose, since related viruses are mostly antagonistic and they compete for similar ecological niches [12]. “Going alone” or “going together” are both adaptive strategies [38], but when co-existing with related viruses in the same host, for viruses like orthotospoviruses, “going alone” may be a better choice even if their vector prefers mixed-infected plants as host. In our study, all results support an aptitude of *F. occidentalis* to retain and transmit (Figure 3) only one virus (preferentially INSV) from mixed-infected plants. We suspect that orthotospoviruses achieve “going alone” by competing for binding sites on their insect vector midgut or by competing for resources for replication, since few of the thrips that fed on mixed-infected plants had both viruses present in their body at two days after adulthood. TSWV glycoproteins (Gn) found on the surface of the virions bind to thrips midgut cells [61], and feeding thrips with a soluble form of Gn can significantly reduce virus titer and inhibit transmission [61,62,63]. However, what could be the ecological advantage for INSV and TSWV in “going alone”, when “going together” would allow them to generate new recombinants and reassortants? While, in our study, INSV was more competitive than TSWV, we speculate that this result is more strain- than species-dependent, and that, in other cases, more competitive strains of TSWV could outcompete weaker strains of INSV.

Complex interactions can happen during mixed infection such as virus–virus, virus–host, and virus–vector interactions. Little is known about how virus–host–vector interactions change when there is more than one virus infecting the plants and/or more than one virus transmitted by the vector, and how that could influence virus evolution in the long run.

## 4. Materials and Methods

### 4.1. Thrips Maintenance

*F. occidentalis* (colony was initially collected from Penn State University Headhouse, 2014) were reared on green bean pods (*Phaseolus vulgaris*) as described previously [64]. Green bean pods were exposed to females for oviposition for 24 h and moved to new Tupperware to synchronize thrips life stage. Synchronized adult females were used in the experiments.

### 4.2. Orthotospoviruses and Their Maintenance

TSWV PA01 isolate and INSV UP01 isolate were used in this study [65,66]. *Nicotiana benthamiana* plants at leaf stage 3–4 were mechanically inoculated with virus-infected tissue stored at −80 °C. *E. sonchifolia* plants were mechanically inoculated with fresh tissue collected from healthy, TSWV-infected, or INSV-infected *N. benthamiana.*

To produce mixed infection, *E. sonchifolia* plants were inoculated with a mixture of TSWV-and INSV-infected *N. benthamiana* fresh tissue in a 5:2 ratio (*w/w*), as this ratio gave a constant number of plants infected by both viruses. 

All plants were maintained in growth chambers at 25 °C with a 16 h light/dark photoperiod for symptom development. Inoculated *E. sonchifolia* plants were tested by ELISA to ascertain their infectious status using the following procedure: leaves were ground in Phosphate Buffered Saline-Tween (PBST) buffer (8.0 g of sodium chloride, 1.15 g of sodium phosphate dibasic, 0.2 g of potassium phosphate monobasic, 0.2 g of potassium chloride, and 0.5 g of Tween-20 dissolved in 1 L of H_2_O, pH 7.4) and subjected to ELISA tests using Agdia (Elkhart, IN, USA) kits following the recommended protocol, in technical duplicates. If the average of the duplicate wells was over three times the absorbance measurement of the negative control, then that sample was considered to be positive. We used mixed-infected plants with similar O.D. reads for each virus (O.D. around 1.7) for the following experiments. ELISA was used to determine the approximate number of transmissible units in the infected plants, instead of the number of genomic segments measurable by qPCR. Using commercial antibodies allowed establishing that the same number of virions was present between single and mixed infection for each virus species, but there could be discrepancies between the number of particles of INSV and TSWV.

### 4.3. Dual Choice Tests

Groups of 20 adult female thrips were starved for 24 h and then allowed to choose between leaf discs of *E. sonchifolia* plants infected with mixed-infected vs. mock-inoculated, mixed-infected vs. TSWV-infected, or mixed-infected vs. INSV-infected. Prior to each test, *E. sonchifolia* plants were tested by ELISA as reported above. For each pairwise comparison, the test was repeated 18 times. The paired leaf discs of 11 mm in diameter were punched from mature *E. sonchifolia* leaves at a similar position and placed on a thin layer of 1% water agar in a 60 × 15 mm Petri dish. A small piece of Kimwipes (Kimberly-Clark Professional Kimtech Science) was placed in the center of the Petri dish. Twenty female adults were shortly anesthetized on ice and then placed on the Kimwipes [67]. The Petri dishes were sealed with parafilm and placed at room temperature under constant light. After 24 h, adult female thrips were removed. Leaf discs were stained and the number of eggs inside each leaf disc was recorded.

### 4.4. Thrips Egg Staining

A method for staining leafhopper eggs was modified to stain and count eggs embedded in leaf discs by female thrips [57,68]. Leaf discs were removed from the Petri dishes and individually immersed in a glass vial containing 2 mL of McBride’s staining solution (95% ethanol and glacial acetic acid (1:1, *v/v*) with 0.2% acid fuchsin). Vials were placed on a plate shaker at low speed for 8 h at room temperature; then, leaf discs were transferred to clean vials containing a de-staining solution (lactic acid/glycerol/water (1:1:1, *v/v*)). Clean vials containing the de-staining solution and leaf discs were transferred into an incubator at 80 °C for 72 h. The leaf discs were carefully washed with water to remove the de-staining solution and examined under light microscope.

### 4.5. Volatile Collection and Analyses

We collected volatiles from four plants of the same age and size three weeks after mechanical inoculation for each treatment group: INSV-infected, TSWV-infected, mixed-infected, and mock-inoculated (mock: *E. sonchifolia* plants were mock-inoculated with PAUL’s buffer; INSV: INSV-infected plants; TSWV: TSWV-infected plants; mixed: plants that were mixed-infected with TSWV and INSV ). Volatiles were collected using solid-phase microextraction (SPME) fibers from one leaf in the same position and of the same size per plant (still attached). Volatile collections were conducted for 24 h in custom-made 15 × 15 × 1 cm Teflon/glass chambers. One attached symptomatic *E. sonchifolia* leaf was enclosed into each chamber with an SPME fiber (Sigma-Aldrich, St. Louis, MO, USA) located on top of the leaf. After collection, SPME fibers were injected into the injection port of a gas chromatographer and analyzed by mass spectrometry (Agilent 7890 + Series GC, Agilent Technologies, Santa Clara, CA, USA). Data were analyzed by Agilent’s enhanced data analysis software, and compounds were found by chromatogram deconvolution. In order to evaluate whether the plant volatile blends changed with virus infection, we removed common containments and selected volatile compounds that were produced at least twice within any one of the treatments as “candidate compounds” [69]. This selection gave us a manageable number of volatile compounds (42). We used decane to normalize the quantity of volatile compounds for PCA analysis using R [70].

### 4.6. Virus Acquisition and Retention Assay

Newly hatched thrips larvae were collected and placed on mixed-infected detached *E. sonchifolia* leaves enclosed in Petri dishes to acquire viruses for 48 h. After the virus acquisition period, thrips were transferred onto green bean pods placed in small plastic containers until they reached adulthood. Two days after they reached adulthood, five adults thrips were pooled and total RNA was extracted using the Quick-RNA Plant Miniprep Kit from Zymo Research (Zymo Research Corporation: Irvine, CA, USA). Reverse transcription was performed using the High-Capacity cDNA Reverse Transcription kit from Applied Biosystems (ThermoFisher Scientific, Waltham, MA, USA) and random primers. After RT-PCR reactions, cDNAs were diluted with a 1:1 ratio of water followed by PCR using 2 × PCR Super Master Mix (Bimake, Houston, TX, USA) with specific primers (Table 1) designed for each virus to detect the presence of TSWV and INSV.

### 4.7. Transmission Assay

Newly hatched thrips larvae were collected and placed on leaf discs infected with TSWV, INSV, or both viruses for a 24 h acquisition period. Thrips were then transferred and reared to adulthood on green bean pods. Adult thrips were placed individually on *E. sonchifolia* leaf discs contained in 1.5-mL tubes for an inoculation access period of 48 h. After the inoculation access period, *E. sonchifolia* leaf discs were separately floated on water in 24-well plates and incubated for one week. ELISA was used to test virus transmission [71].

### 4.8. Data Analysis

The percentages of eggs for each leaf disc relative to the total number of eggs embedded in each disc (relative attraction) were calculated. Data were analyzed with Wilcoxon signed rank test on the relative attraction of oviposited eggs on each leaf disc pair. Virus retention rate for single thrips was calculated using the pooledBin function in R package binGroup [47]. Virus transmission efficiency was analyzed with Student’s *t*-test.

## 5. Conclusions

A recent study suggested that the evolution of INSV is driven by reassortment with other INSV strains rather than recombination [36], and no proof exists that INSV can reassort with other orthotospoviruses, including TSWV, a virus often found to co-infect INSV-infected plants. Experiments indicated that more closely related bunyaviruses are more likely to produce viable reassortants [32,72]. However, superinfection exclusion can restrict very closely related viruses from establishing mixed infections in plants. The balance between the evolutionary advantage offered by creating reassortants and the competition for resources when viruses occupy the same niche, while keeping genomic stability, allows viruses to maintain an open window for adaptation to a changing environment [30]. Based on these considerations, we could speculate that orthotospoviruses need a narrow bottleneck during transmission in order to keep a certain level of genomic stability, and insect vectors are not the “mixing vessel” for them to reassort and recombine. Our study shows that TSWV and INSV can successfully co-infect the same plant, but vector thrips select which one of the two viruses will be successfully transmitted to the new plant host, supporting the idea that thrips serve as the transmission bottleneck for mixed infection of two *Orthotospoviruses*. This would suggest that plants, instead of insects, are the best host for orthotospoviruses to create new genetic variants, and this may be the reason why vectors are selected to be more attracted to mixed-infected plants. Further long-term studies would be needed to determine if INSV and TSWV can reassort during mixed infections in plants.

## Figures and Tables

**Figure 1 plants-09-00509-f001:**
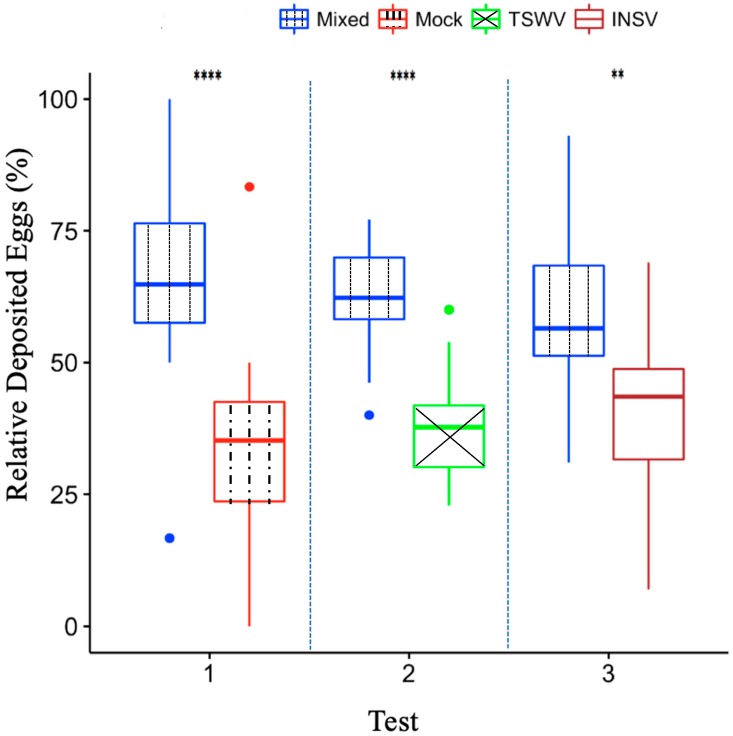
Thrips dual-choice test result. Percentage of eggs deposited during choice tests experiments (relative attraction) in *Emilia sonchifolia*. Choice test: (1) mixed-infected vs. mock-inoculated leaf discs; (2) mixed-infected vs. *Tomato spotted wilt orthotospovirus* (TSWV)-infected leaf discs; (3) mixed-infected vs. *Impatiens necrotic spot orthotospovirus* (INSV)-infected leaf discs (Wilcoxon signed rank test on the relative attraction of oviposited eggs on each leaf disc pair; ** *p* < 0.01, **** *p* < 0.0001; *n* = 18).

**Figure 2 plants-09-00509-f002:**
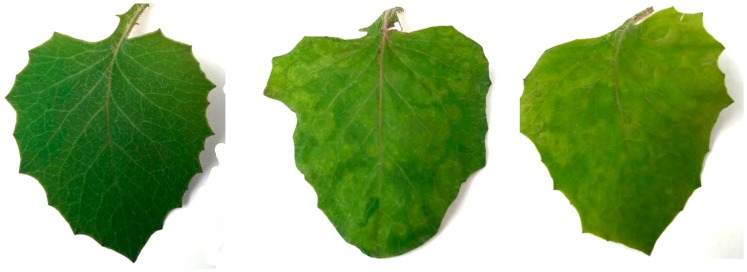
Typical symptoms seen in infected *Emilia sonchifolia* leaves, compared to a healthy leaf. **Left**: healthy leaf; **Middle**: INSV-infected leaf; **Right**: mixed-infected leaf. Infected plants show mosaic and yellow mottling symptoms, not distinguishable in TSWV-, INSV-, or mixed-infected plants. Plants also show similar size.

**Figure 3 plants-09-00509-f003:**
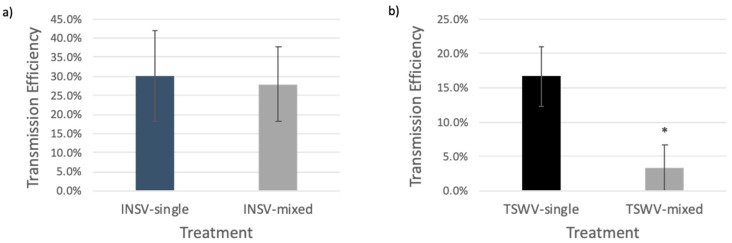
Transmission efficiency of INSV and TSWV from systemically infected *Emilia sonchifolia* leaves. Thrips were collected and placed on leaf discs infected with TSWV, INSV, or TSWV + INSV for 24 h of acquisition period. Thrips were then transferred and reared to adulthood on green bean pods and were given an inoculation access period of 48 h to uninfected leaf discs. Experiments were conducted three times. Individual leaf discs were tested by ELISA for virus infection seven days after transmission. (**a**) Transmission efficiency of INSV from singly and mixed-infected plants; (**b**) transmission efficiency of TSWV from singly and mixed-infected plants. Error bars represent standard deviation. Asterisks indicate a significant difference for TSWV transmission efficiency (*p* < 0.05) using Student’s *t*-test.

**Table 1 plants-09-00509-t001:** List of primers in PCR for virus detection.

Primer	Sequence (5′–3′)
TSWV-L-Forward	TCTCCACCTCGCTTCTTTGT
TSWV-L-Reverse	AAACAAAGGGATGGCAACTG
INSV-L-Forward	AGAGAGGACCACCCTTGGAT
INSV-L-Reverse	ATGTTCGGTGAGCTGGTTTC

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
