# Peer review of "Thrips as the Transmission Bottleneck for Mixed Infection of Two Orthotospoviruses"

_plants, 2020, doi:10.3390/plants9040509_

Round 1

Reviewer 1 Report

In this paper, the authors analyzed thrips vector preference and competency to transmit tomato spotted wilt virus and impatiens necrotic spot virus singly and in combination. The paper is very well-written, and the authors have performed a thorough analysis of their data. I believe that this paper increases our fundamental knowledge of orthotospovirus-thrips interaction specifically in terms of consequences of mixed infection. It is indeed interesting that no natural reassortants have been found and this study sheds light on the potential mechanism. A well-done literature review in the Introduction section and rationale for the study.

One concern that I have is the all preference assays were conducted using leaf discs and I’m sure the authors are aware that leaf damage can affect insect preference. I would suggest they provide some explanation to validate their results. I would suggest changing the colors on the box plots because the differences are not visible in black and white.

I would be cautious over-reaching with oviposition data because number of eggs is not equal to hatching success or number of offspring.

While the volatile analysis is interesting it takes away from the main goal of this study which was to determine how thrips ability to transmit affects mixed infections. The authors state that similar level of virus titer was maintained in host plants used for acquisition assays. I’m curious how the authors did this and how it was confirmed. I did not see any qPCR data on virus titer in the plants. This is a major point of consideration because virus acquisition efficiency has been shown to be related to virus titer in the host plant.

I would remove Fig. 3 as it is not very informative.

The transmission efficiency experiment results are interesting. I would have expected TSWV to be transmitted at a higher rate and indeed it is more prevalent than INSV. It would be helpful to provide some explanation for this result.

It would be helpful if the authors had done long-term experiments to determine any reassortants in mixed infections and to see if the dynamics of TSWV and INSV mixed infections changes over time in whole plants.

Author Response

Thank you and all the reviewers for the constructive suggestions given, we hope we addressed all of them and we feel that our paper has improved by incorporating the suggested changes. We also extensively reviewed the paper for grammar and syntax. All changes are outlined as track changes in the document and the major ones are reported below here.

-One concern that I have is the all preference assays were conducted using leaf discs and I’m sure the authors are aware that leaf damage can affect insect preference. I would suggest they provide some explanation to validate their results.

This observation is absolutely valid, and we are aware that leaf damage can affect insect preference. We added a note of caution in the discussion and added references of other publications where leaf disks were used and compared to the use of whole plants.

-I would suggest changing the colors on the box plots because the differences are not visible in black and white.

We modified the box plot (Figure 1) for better presentation and added patterns for color blind readers. But since this journal accept figure with colors, we would like to keep current color schemes if possible.

-I would be cautious over-reaching with oviposition data because number of eggs is not equal to hatching success or number of offspring.

We agree and added a note in the discussion.

-The authors state that similar level of virus titer was maintained in host plants used for acquisition assays. I’m curious how the authors did this and how it was confirmed. I did not see any qPCR data on virus titer in the plants. This is a major point of consideration because virus acquisition efficiency has been shown to be related to virus titer in the host plant.

We did not use qPCR to quantified virus titer due to the study by Schreur, P. J. W., & Kortekaas, J. (2016). (Single-molecule FISH reveals non-selective packaging of Rift Valley fever virus genome segments. PLoS pathogens, 12(8).) This study showed that a cell could contain different amounts of each RNA segment for bunyavirus and we don’t know which segment is the best indicator of virus titer. Therefore, we used ELISA to quantify virus titer of infected plants since the antigen (structural protein) could represent better what is acquired by the thrips.

Even if all treatments had an ELISA OD around 1.7, we cannot ensure that amount of particles of INSV and TSWV was the same, since we used commercial antibodies, but we can for sure state that the level of each virus was the same between single and double infection. We added the ELISA data in the material and method and an explanation in the discussion.

-I would remove Fig. 3 as it is not very informative.

The original Fig 3 has been removed

-The transmission efficiency experiment results are interesting. I would have expected TSWV to be transmitted at a higher rate and indeed it is more prevalent than INSV. It would be helpful to provide some explanation for this result.

The transmission rate of TSWV and INSV from single virus infected plants did not show statistic differences (P > 0.05). But in mixed infected plants, the transmission rate of INSV is much higher than TSWV. We suspect this could due to direct competition for binding sites in thrips midgut or by competition of host resources during replication, since few of the thrips that fed on mixed infected plants had both viruses present in their body at 2 days after adulthood. These notes are added in the paper in the discussion.

We heard from other researchers that different strains of INSV and TSWV can be more competitive, and our results could depend on the ‘strength’ of our INSV strain. We added a clarification in the discussion. 

-It would be helpful if the authors had done long-term experiments to determine any reassortants in mixed infections and to see if the dynamics of TSWV and INSV mixed infections changes over time in whole plants.

Yes, that’s a great suggestion. We would like to include this experiment into our future work plan. In order to generate reassortants, two viruses need to be present in the same cell. We are also interested to look at virus localization at the single cell level in our future work. We added this point in the conclusions.

Reviewer 2 Report

The paper entitled "Thrips as the transmission bottleneck for mixed-infection of two Orthotospoviruses" presents the results of several experiments that provide insight into the interaction between two persistent propagative viral plant pathogens.

Introduction: The authors have a good balance between citing all relevant literature and overall length. Beyond some minor editorial changes and the inclusion of results/discussion in the final paragraph the introduction is good. 

Results: The first paragraph should be in the introduction. It is not a result because the authors did not generate those data. There are other locations where text belongs in other sections. The figure legends need to be improved. Sometimes they leave out important detail, while other times they read like methods in disguise.

   I suggest the authors look at the R package binGroup to improve the analysis of infection data. 

install.packages("binGroup")
library(binGroup)
#n=number of pools, m=pool size, x=infected
pooledBin(x=137, m = 1, n =184)

My initial impression of Figure 3 was favorable, but then I could not figure out how to interpret the figure. It looks like a Venn diagram but there are problems. If it is only random shapes then it could be removed or replaced.

The lone asterisk in Figure 4 might be confusing. Letters across the top of the bar chart would make it clearer that m-TSWV is significantly different from all other treatments and there are no significant differences between the other treatments. 

Methods: This section needs more detail and a bit of reorganization.

Overall I think the experiment was interesting and it is easy to understand what was done, why it was done, and the outcome. The experimental design was appropriate. It has enough depth to act as a starting point for additional research by this group or others. It is worth publishing and this journal is appropriate.

Author Response

Thank you and all the reviewers for the constructive suggestions given, we hope we addressed all of them and we feel that our paper has improved by incorporating the suggested changes. We also extensively reviewed the paper for grammar and syntax. All changes are outlined as track changes in the document and the major ones are reported below here.

-Results: The first paragraph should be in the introduction. It is not a result because the authors did not generate those data.

We moved the first paragraph from the results to the introduction.

-The figure legends need to be improved. Sometimes they leave out important detail, while other times they read like methods in disguise.

We improved the figure legends as well as the figures. Please check the revised Figure 1-3. We hope we addressed all concerns.

-I suggest the authors look at the R package binGroup to improve the analysis of infection data.

This is a great suggestion, thank you! We calculated the virus retention rate using pooledBin function and added the results in section 2.2.

-My initial impression of Figure 3 was favorable, but then I could not figure out how to interpret the figure. It looks like a Venn diagram but there are problems. If it is only random shapes then it could be removed or replaced.

The original Fig 3 has been removed

-The lone asterisk in Figure 4 might be confusing. Letters across the top of the bar chart would make it clearer that m-TSWV is significantly different from all other treatments and there are no significant differences between the other treatments.

We remade the original Figure 4 (now Figure 3) for better representation and more appropriate analysis

-Methods: This section needs more detail and a bit of reorganization.

We reorganized and added more detail throughout the Material and Methods section.

Reviewer 3 Report

- A bottleneck implies a restriction of some kind, but it needs to be defined with supported experiments here.

- The last sentence in the abstract is meaningless - what is ‘viral fitness’?

- There seems to be no conclusion to back up the title.

Author Response

Thank you and all the reviewers for the constructive suggestions given, we hope we addressed all of them and we feel that our paper has improved by incorporating the suggested changes. We also extensively reviewed the paper for grammar and syntax. All changes are outlined as track changes in the document and the major ones are reported below here.

-A bottleneck implies a restriction of some kind, but it needs to be defined with supported experiments here.

We are not sure if we interpreted correctly this observation, we added extra explanations in the manuscript, and we hope that we addressed this observation in a sufficient manner. Our data also suggest that there is a restriction during transmission, since transmission of both INSV and TSWV was never observed in our experiments.

-The last sentence in the abstract is meaningless - what is ‘viral fitness’?

Replication and transmission contribute to the prevalence of genetic material at population level in the field over time. Therefore, overall viral fitness includes replication, transmission and epidemiologic fitness. We added a better and more complete explanation for what we mean with viral fitness according to published literature in the abstract and introduction.

-There seems to be no conclusion to back up the title.

Thank you! we added a paragraph in the conclusions to back up the title.

Round 2

Reviewer 1 Report

I read the revised manuscript and the author's comments. I am satisfied and have no other changes. Thanks,

Reviewer 3 Report

Hi editor

The manuscript has been improved, however, it's English still needs further work. For example, in line 297 "USE "was used twice in a wrong format.